# Diagnostic potential of total serum ghrelin in autoimmune gastritis: A systematic review and meta-analysis

Iqbal Taufiqqurrachman[1,2], Andro Pramana Witarto [ID][3,4], Ari Fahrial Syam[5], Murdani Abdullah[5,6], Sigit Ari Saputro[7,8], Irine Normalina[9], Muhammad Miftahussurur [ID][9,10]*, Yoshio Yamaoka [ID][1,9,11]*

1 Department of Environmental and Preventive Medicine, Oita University Faculty of Medicine, Yufu, Japan, 2 Faculty of Medicine, Universitas Indonesia, Jakarta, Indonesia, 3 Internal Medicine Specialist Study Program, Department of Internal Medicine, Faculty of Medicine, Universitas Airlangga, Surabaya, Indonesia, 4 Internal Medicine Specialist Study Program, Department of Internal Medicine, Dr. Soetomo General Academic Hospital, Surabaya, Indonesia, 5 Division of Gastroenterology, Department of Internal Medicine, Faculty of Medicine, Universitas Indonesia, Cipto Mangunkusumo General Hospital, Central Jakarta, Indonesia, 6 Human Cancer Research Center, Indonesian Medical and Research Education Institute, Faculty of Medicine, Universitas Indonesia, Jakarta, 7 Department of Epidemiology, Biostatistics, Population and Health Promotion, Faculty of Public Health, Airlangga University, Surabaya, East Java, Indonesia, 8 Center of Excellence for Patient Safety and Quality, Universitas Airlangga, Surabaya, East Java, Indonesia, 9 Helicobacter pylori and Microbiota Study Group, Institute Tropical Disease, Universitas Airlangga, 10 Division of Gastroentero-Hepatology, Department of Internal Medicine, Faculty of Medicine-Dr. Soetomo Teaching Hospital, Surabaya, Indonesia, 11 Division of Genome-Wide Microbiology, Research Center for Global and Local Infectious Diseases (RCGLID), Oita University, Yufu, Oita, Japan

* yyamaoka@oita-u.ac.jp (YY); muhammad-m@fk.unair.ac.id (MM)

## Abstract

### Background

Autoimmune gastritis (AIG) is a chronic inflammatory condition characterized by the destruction of gastric parietal cells. The invasive nature of diagnostic procedures and risk of confounding factors hinder the development of reliable diagnostic tools for AIG.

### Methods

We conducted a systematic search of four databases. After assessing the study quality using the ROBINS-E tool, a meta-analysis was performed using a random-effects model. Meta-regression and sensitivity analyses were conducted to explore the sources of heterogeneity and impact of study bias,

### Results

The pooled mean difference in total serum ghrelin (pmol/L) between patients with AIG and healthy controls was −65.28 (95% CI: −178.54, 47.97). Subgroup analysis showed that the mean differences in serum ghrelin for mild, moderate, and severe atrophy were −78.85 (95% CI: −165.17, to 7.48), −91.97 (95% CI: −183.11, to −0.84), and −110.67 (95% CI: −204.77, to −16.56), respectively. The sensitivity

**Data availability statement:** All relevant data are within the paper and its Supporting Information files.

**Funding:** This report is based on work supported in part by grants from the National Institutes of Health (DK62813) (Y.Y) and Grants-in-Aid for Scientific Research from the Ministry of Education, Culture, Sports, Science, and Technology (MEXT) of Japan (221S0002, 21H00346, 22H02871, and 23K24133) (Y.Y). This work was supported in part by the Japan Agency for Medical Research and Development (AMED) [Adopting Sustainable Partnerships for Innovative Research Ecosystem (ASPIRE); 23836904, Science and Technology Research Partnership for Sustainable Development (SATREPS); 21357105] and the Japan International Cooperation Agency (JICA) [SATREPS] (Y.Y). IT is a Ph.D. student supported by the Japanese Government (MEXT) Scholarship Program for 2024. the funders had no role in study design, data collection and analysis, decision to publish, or preparation of the manuscript.

**Competing interests:** The authors have declared that no competing interests exist.

analysis confirmed that the exclusion of studies with high-risk bias did not significantly alter the results. Meta-regression indicated that BMI contributed substantially to heterogeneity.

## Conclusions

Although total serum ghrelin levels were not significantly different between patients with AIG and healthy controls, significantly lower levels were observed in patients with moderate-to-severe gastric atrophy. Given the high heterogeneity and limitations of existing studies, the diagnostic utility of serum ghrelin in AIG warrants further investigation.

## Introduction

Autoimmune gastritis (AIG) is a chronic gastric inflammation that is caused by the destruction of the subunit $H^+/K^+ATPase$ pump in parietal cells induced by autoreactive T helper 1 cells after the production of anti-parietal cell antibodies (APCA) or anti-intrinsic factor antibodies (AIFA) [1–3]. It is characterized by the damage of oxyntic mucosa or parietal cells at the fundus and corpus of the stomach, and can lead to progressive mucosal atrophy [1,4–6], therefore, it is classified within the scope of chronic atrophic gastritis. The prevalence of AIG is heterogeneous and depends on diagnostic criteria. In the general population, the prevalence of AIG is 0.1%–2.7% if the diagnosis is based on esophagogastroduodenoscopy (EGD) and histopathological examination, but the prevalence increases to 8%–20% if the diagnosis is based on the positive detection of APCA or AIFA [7,8].

Historically, the incidence of AIG in Asian countries was lower than that in Western countries, as chronic atrophic gastritis in Asia has primarily been attributed to *Helicobacter pylori* infection. A report in Japan showed that from 6,739 subjects, there are 46 (0.49%) had the endoscopic appearance of AIG, but only 33 of 46 were diagnosed as AIG [9]. Another report evaluates the AIG appearance in biopsy sample of several ethnicity, and the results showed that the AIG appearance only found in 1.4% of Asian ethnicity [10]. A recent meta-analysis showed that the Europe, Africa, and Australia region were dominating the prevalence of AIG with the prevalence of 4.94%, 8.46%, and 8.08%. In comparison, the prevalence of AIG in Asia region is only 2.23% [11]. However, recent studies suggest a shifting epidemiological trend, with increasing recognition of AIG among Asian populations.

The prior knowledge of low prevalence of AIG in Asian population may be masked by the high rate of *H. pylori* infection, difficulty in endoscopic diagnosis based on atrophic appearance that can misdiagnose *H. pylori* infection, and false-positive urea breath test (UBT) results due to the increase in urease-positive, non-*H. pylori*, bacterial growth induced by achlorydia [12,13]. A supporting report showed that the eradication of *H. pylori* was followed by the rapid progression of AIG, indicating that *H. pylori* infection obscures AIG manifestations [12].

This has changed our paradigm regarding the occurrence of AIG, particularly in Asian countries. Therefore, a prompt diagnostic modality is required for AIG.

Previous studies have used endoscopic appearance, biopsy examination, and positive APCA or intrinsic factor autoantibodies results to diagnose AIG [14]. However, because endoscopy and biopsy are invasive and it is difficult to differentiate between AIG and prior *H. pylori* infection [15], and several factors are associated with APCA positivity [16], an alternative non-invasive diagnostic method for AIG with high sensitivity and specificity is needed.

Ghrelin is a potential biomarker of AIG. Ghrelin is a hormone secreted by endocrine cells in the oxyntic glands of the gastric corpus and the fundus, particularly by the P/D1 cells. This hormone has been reported to be associated with several conditions such as gastric cancer and colorectal cancer. In AIG, damage to the oxyntic mucosa or parietal cells at the fundus and corpus of the stomach may lead to progressive gastric mucosal atrophy. The atrophy may affect the level of ghrelin that predominantly produced in stomach [17,18]. (**Fig 1**)

However, the general difference in serum ghrelin levels between patients with AIG and healthy individuals is unknown, although a few studies have used ghrelin in diagnostic studies of AIG. This study aimed to determine the differences in serum ghrelin levels and corresponding differences in the severity of gastric atrophy between adult patients with AIG and healthy controls.

## Methods

### Search strategy

A systematic review and meta-analysis was performed based on the PRISMA 2020 checklist [19]. Articles were sourced from four different databases: (1) PubMed, (2) EBSCO Host in MEDLINE Full Text, (3) Scopus, and (4) ProQuest. The search period was August 25 to September 30, 2024. In this study, the population comprised adult patients with normal exposure to autoimmune gastritis (with terms for the search of "Atrophic Gastritis" or "Autoimmune Gastritis" or "Atrophic Autoimmune Gastritis"), and the outcome was total serum ghrelin (with terms for the search of "Ghrelin" or "Serum Ghrelin"). The search strategy used to find relevant articles in this study was based on different databases and registers using the keywords listed in S1 Table.

### Eligibility criteria

The inclusion criteria were as follows: (1) adult patients (> 18 years old) and (2) diagnosis of autoimmune gastritis based on gastric atrophic appearance (from EGD or histological examination) and positive APCA with or without *H. pylori*

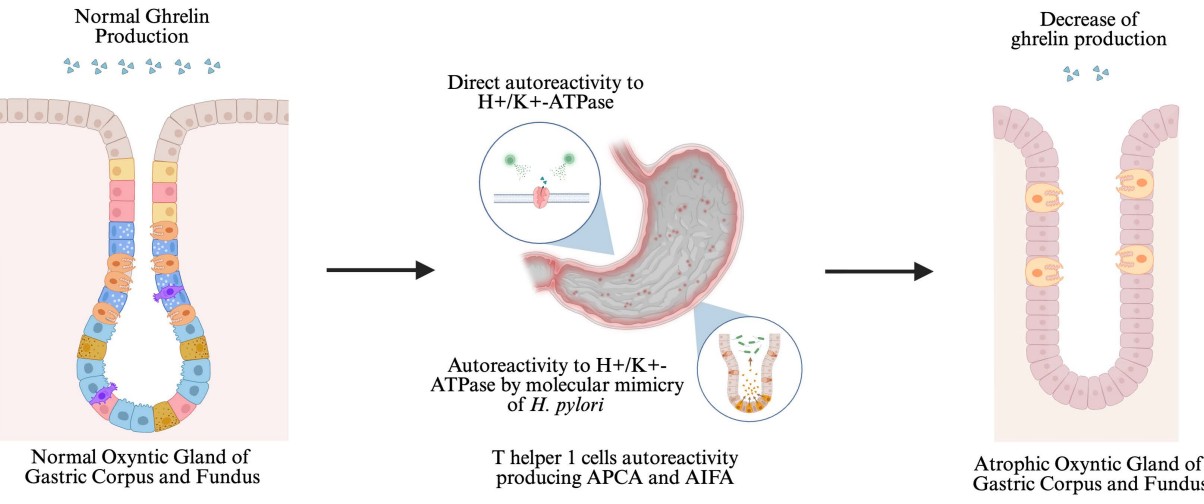

**Fig 1. Relationship between autoimmune gastritis, ghrelin, and gastric atrophy.**

infection. This criteria was based on the recent finding that AIG could be triggered by *H. pylori* infection [20]. The exclusion criteria were as follows: (1) history or status of malignancy and its treatment; (2) renal impairment; (3) other conditions that cause gastric inflammation (e.g., prolonged non-steroid anti-inflammatory drug use, prolonged corticosteroid use, and alcohol abuse); (4) history of gastric surgery; and (5) other conditions or procedures that obscure the gastric appearance on EGD.

### Selection and data collection process

Studies were progressively selected. Two reviewers considered articles and independently selected relevant studies using a progressive approach (relevance of the title and abstract, deduplication, and full text) based on the eligibility criteria. The total number of full-text relevant studies was compared with each reviewer's input on a Google sheet spreadsheet. If perceptions differed among the reviewers, discussions and voting were conducted by all five reviewers. Data from the studies were collected from the metadata of the Google spreadsheets. Data collection was performed by two reviewers. No automated or artificial intelligence tools were employed for collecting data from the selected studies. If conflict existed between the two reviewers, discussions and voting were conducted among all five reviewers.

The outcome data of the selected studies were the mean total serum ghrelin levels between two different groups (patients with autoimmune gastritis and healthy controls) and different degrees of atrophic severity. Additional data sought from the selected studies included the number of subjects (total, autoimmune gastritis, normal, and atrophic severity), demographic status of the subjects (age, sex, and body mass index), and the method used to assess serum ghrelin levels.

### Risk of bias assessment

The risk of bias in the systematic reviews of observational studies was assessed with the Risk of Bias in Non-Randomized Studies – Of Exposure (ROBINS-E) tool. The components assessed using this tool were as follows: (1) bias due to confounding factors, (2) bias arising from measurement of the exposure, (3) bias in selection of participants into the study (or into the analysis), (4) bias due to post-exposure interventions, (5) bias due to missing data, (6) bias arising from measurement of the outcome, and (7) bias in selection of the reported result, to be accumulated as the overall bias [21]. To evaluate the robustness of the meta-analysis result, a sensitivity analysis will be performed if the included studies had a high risk of bias.

### Data synthesis

Data were generated based on the demographic status of the subjects, followed by the collection of effect measures, which included the mean total serum ghrelin (in pmol/L or pg/mL) of AIG patients and healthy controls, and the mean total serum ghrelin level between AIG patients with different atrophic severities and healthy controls. Unavailable data on mean total serum ghrelin levels were reported as N/A (not available) and were not included in the data analysis. The mean total serum ghrelin levels were converted to the pmol/L scale by multiplying the value in pg/mL by a coefficient of 0.3, based on the findings of Aukan *et al*. (2022) [22]. The generated data are presented in Table 1. Meta-analysis was performed for continuous data with a random effects model using RStudio version 2024.09.0+375 (© Posit Software, PBC) to collect data on effect size, variance, heterogeneity, and publication bias, which were presented in forest and funnel plots. Funnel plot asymmetry was measured using Egger's regression test if > 10 studies were included ($k > 10$). When heterogeneity was high, meta-regression was performed using STATA version 17.0 (StataCorp).

### Ethics approval and consent to participate

Ethical approval was not required.

**Table 1. Study characteristics.**

| Author (Year), Country | Subjects' Characteristics | Number of Subjects | Method of Diagnosis for AIG | Method of Serum Ghrelin Measurement | Mean±SD of Total Serum Ghrelin in pmol/L | | Limitation of the Study |
|---|---|---|---|---|---|---|---|
| | | | | | AIG | HC | |
| Panarese, et al. [23] (2020) - Italy | **Inclusion criteria:** Patient with dyspepsia who undergoing the endoscopy or positive APCA. **Exclusion criteria:** Had significant comorbidities (cancer, thyroid, liver, or renal disease), prior gastric surgery, recent use of proton pump inhibitors or antibiotics within one month. | AIG (18) HC (18) | EGD Histopathology (OLGA atrophic scoring system) APCA positive | ELISA | 67.83±72.90 | 43.38±33.30 | No restriction of BMI across group or during analysis |
| Checchi, et al. [26] (2007) - Italy | **Inclusion criteria:** Patient with autoimmune thyroiditis and positive APCA who screened for autoimmune gastritis. **Exclusion criteria:** Recent use of drugs affecting gastric function, prior gastric surgery, malignancy, chronic systemic illness, alcohol abuse, BMI > 30 kg/m$^2$ | AIG (40) HC (40) | EGD Histopathology (Sydney system classification) APCA positive | Radio immuno-assay | 158.80±8.79 | 331.47±41.15 | Only 52 of 233 subjects who underwent endoscopy for histologic confirmation; No stratification of *H. pylori* status |
| Campana, et al. [24] (2007) - Italy | **Inclusion criteria:** Patient with histologically confirmed of chronic atrophic gastritis involving corpus/fundus with BMI < 25 kg/m$^2$. **Exclusion criteria:** Had *H. pylori* infection, diabetes mellitus, hypertension. | AIG (25) HC (25) | EGD Histopathology | Radio immuno-assay | 195.00±90.00 | 255.00±134.00 | No subgroup analysis for each gastric atrophy severity |
| Alonso, et al. [25] (2007) - Spain | **Inclusion criteria:** Patient with T1DM with or without autoimmune atrophic gastritis, matched with healthy controls for age, sex, and BMI. **Exclusion criteria:** Had renal impairment, infection, liver disease or history of *H. pylori* eradication. | AIG (15) HC (30) | EGD Histopathology APCA positive | Radio immuno-assay | 195.30±84.6 | 177.90±60.81 | No subgroup analysis for each gastric atrophy severity; Had confounding variables (the control of T1DM and insulin dose) |
| Gao, et al. [27] (2008) - China | **Inclusion criteria:** Patient aged > 65 years who underwent endoscopy for dyspepsia, matched with healthy controls for age, sex and BMI. **Exclusion criteria:** Had BMI < 18.5 or > 25 kg/m$^2$, had diabetes, systemic illness, *H. pylori* infection, recent medications, alcohol abuse, and prior surgery. | AIG (50) HC (50) | EGD Histopathology (Sydney system classification) *H. pylori* negative | Radio immuno-assay | 108.96±11.45 | 215.78±19.45 | N/A |

AIG, autoimmune gastritis. HC, healthy control. CAG, chronic autoimmune gastritis. BMI, body mass index. T1DM, type 1 diabetes mellitus. T2DM, type 2 diabetes mellitus.

EGD, esophagogastroduodenoscopy. APCA, anti-parietal cell antibodies. *H. pylori*, *Helicobacter pylori*. SD, standard deviation.

## Results

### Study selection

Study selection was performed using a progressive approach from four databases (PubMed, EBSCO Host in MEDLINE full text, Scopus, and ProQuest). The PRISMA flow [19] is shown in Fig 2.

### Study characteristics

The study's characteristics are presented in **Table 1**. The five included studies comprised a total of 331 participants across Italy, Spain and China with sample sizes ranging from 15 to 100 subjects. All studies diagnosed AIG through a combination of endoscopic evaluation, histopathological confirmation, and APCA positivity, while methods for serum ghrelin

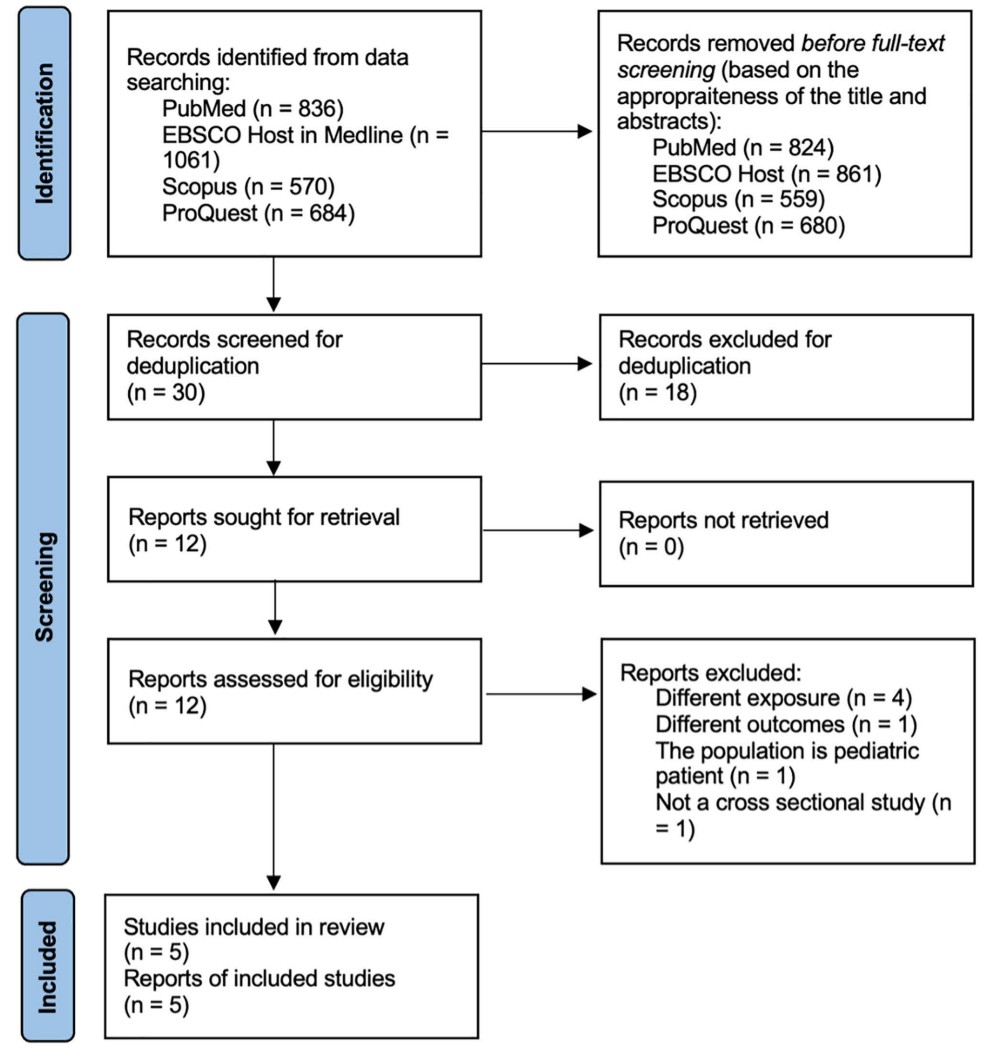

**Fig 2. Study selection process.**

measurement varied between ELISA and radioimmunoassay. Participant demographic differed across studies, reflecting differences in diagnostic methods, population characteristics, and sample handling. Key study limitations included lack of BMI stratification, incomplete *H. pylori* status reporting, absence of atrophy severity subgroup analyses, and potential confounding from comorbid conditions such as type 1 diabetes mellitus.

### Risk of bias in studies

The risk of bias was measured using ROBINS-E. Two studies showed a high risk of bias (Panarese, et al. (2007) and Checchi, et al. (2007)), one study had some concern regarding bias (Alonso, et al.), and two studies showed a low risk of bias (Alonso, et al. (2007) and Gao, et al. (2008)) (**Fig 3**).

### Data synthesis

**Difference in total serum ghrelin between autoimmune gastritis patients and healthy controls.** The mean difference in total serum ghrelin (pmol/L) between autoimmune gastritis and healthy controls from five studies (311

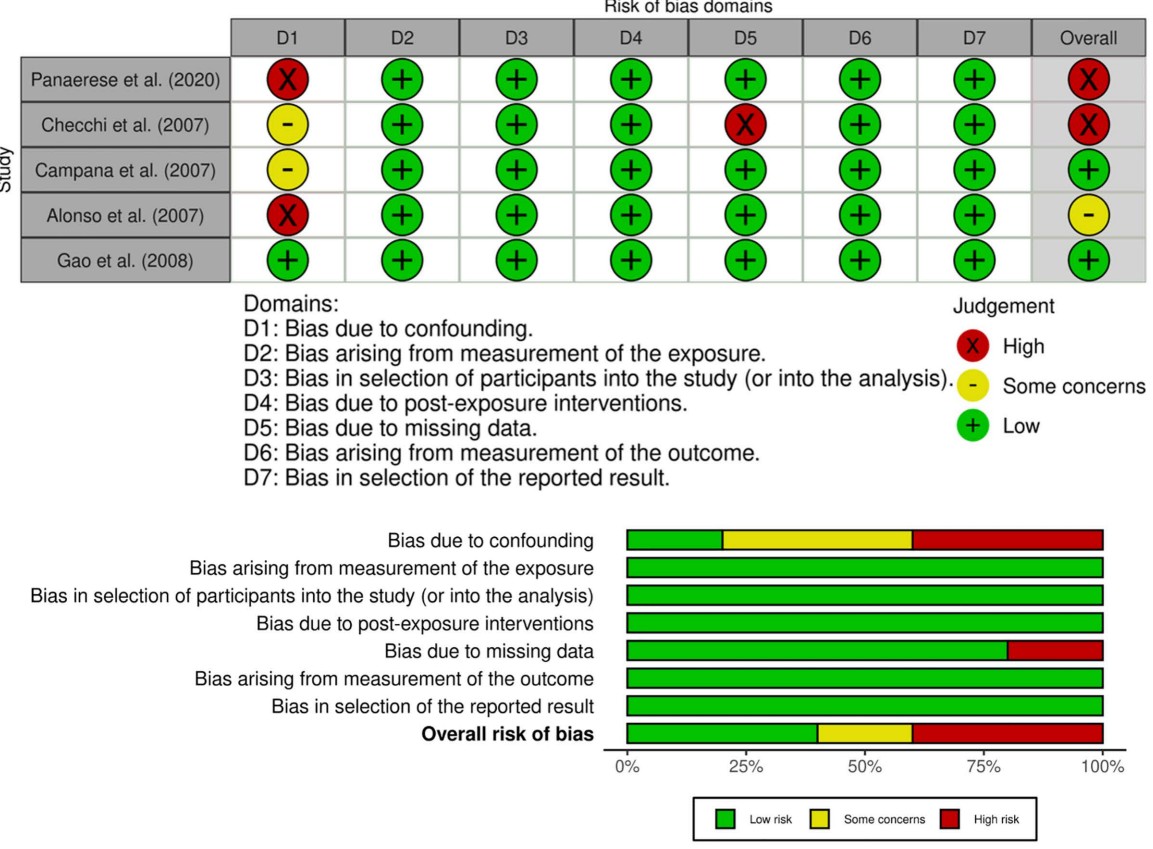

**Fig 3. Risk of Bias Assessment.**

subjects) was –65.28 (95% confidence interval [CI]: –178.54, 47.97) pmol/L (Fig 4). However, the heterogeneity of the results was very high, with an $I^2$ value of 98%.

**Difference in total serum ghrelin between AIG patient with different degrees of gastric atrophy to healthy controls.** The mean differences between different degrees of atrophy were analyzed. The resulting forest and funnel plots are shown in Fig 5. The mean differences in total serum ghrelin (pmol/L) for three different degrees of atrophy (mild vs healthy control, moderate vs healthy control, and severe vs healthy control) were –78.85 (three studies of a total of 189 subjects; 95% CI: –165.17, 7.48), –91.97 (three studies of a total of 181 subjects; 95% CI: –183.11, –0.84), and –110.67 (three studies of a total of 153 subjects; 95% CI: –204.77, –16.56), respectively. These results were accompanied by very high heterogeneity.

### Sensitivity analysis and meta-regression

Sensitivity analysis was performed to assess the precision of the findings, because there are two studies with a high risk of bias were included in the meta-analysis. Sensitivity analysis was performed by excluding high-risk bias studies and evaluating pairwise meta-analyses. The results showed that the mean difference in serum ghrelin levels between patients with AIG and healthy controls was also not significantly different, with a mean difference of −52.51 (95% CI: −126.34; 21.30) (S1 Fig).

To evaluate the possible causes of high heterogeneity, a meta-regression analysis was performed. Two demographic statuses (mean age and BMI) were analyzed using a meta-regression. The results showed that mean age was not associated with high heterogeneity (S2 Table); however, mean BMI was significantly associated with high heterogeneity (S3 Table).

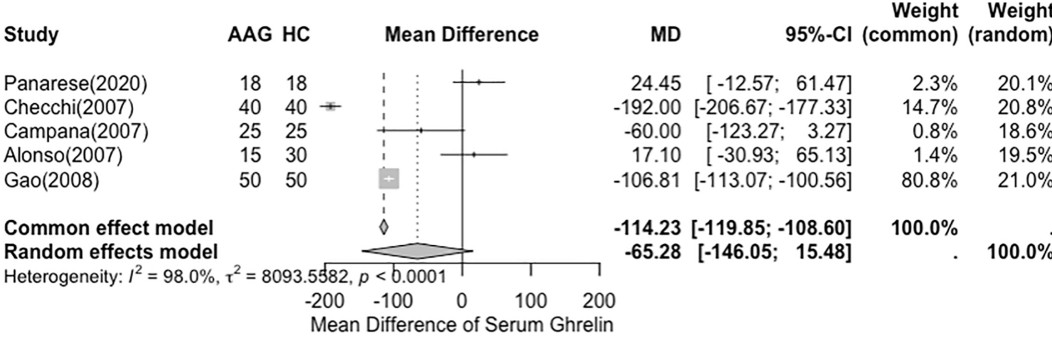

| Study | AAG | HC | Mean Difference | MD | 95%-CI | Weight (common) | Weight (random) |
|---|---|---|---|---|---|---|---|
| Panarese(2020) | 18 | 18 | | 24.45 | [ -12.57; 61.47] | 2.3% | 20.1% |
| Checchi(2007) | 40 | 40 | | -192.00 | [-206.67; -177.33] | 14.7% | 20.8% |
| Campana(2007) | 25 | 25 | | -60.00 | [-123.27; 3.27] | 0.8% | 18.6% |
| Alonso(2007) | 15 | 30 | | 17.10 | [ -30.93; 65.13] | 1.4% | 19.5% |
| Gao(2008) | 50 | 50 | | -106.81 | [-113.07; -100.56] | 80.8% | 21.0% |
| Common effect model | | | | -114.23 | [-119.85; -108.60] | 100.0% | . |
| Random effects model | | | | -65.28 | [-146.05; 15.48] | . | 100.0% |

Heterogeneity: $I^2 = 98.0\%$, $\tau^2 = 8093.5582$, $p < 0.0001$

-200  -100  0  100  200
Mean Difference of Serum Ghrelin

**Fig 4. Forest Plot of the Difference in Total Serum Ghrelin of Autoimmune Gastritis and Healthy Control.**

a) Forest Plot of Total Serum Ghrelin Level Difference between AIG with Mild Gastric Atrophy and Healthy Control

| Study | Mild Atrophic | Healthy Control | Mean Difference | MD | 95%-CI | Weight (common) | Weight (random) |
|---|---|---|---|---|---|---|---|
| Panarese | 22 | 18 | | 4.02 | [ -22.65; 30.69] | 5.2% | 33.6% |
| Checchi | 4 | 51 | | -148.00 | [-193.60; -102.40] | 1.8% | 31.7% |
| Gao | 18 | 50 | | -96.05 | [-102.37; -89.72] | 93.0% | 34.7% |
| Common effect model | | | | -91.74 | [ -97.84; -85.64] | 100.0% | . |
| Random effects model | | | | -78.85 | [-165.17; 7.48] | . | 100.0% |

Heterogeneity: $I^2 = 96.5\%$, $\tau^2 = 5582.3560$, $p < 0.0001$

-150  -50 0 50 100 150

b) Forest Plot of Total Serum Ghrelin Level Difference between AIG with Moderate Gastric Atrophy and Healthy Control

| Study | Moderate Atrophic | Healthy Control | Mean Difference | MD | 95%-CI | Weight (common) | Weight (random) |
|---|---|---|---|---|---|---|---|
| Panarese | 19 | 18 | | -5.16 | [ -29.96; 19.64] | 5.0% | 33.4% |
| Checchi | 22 | 51 | | -166.00 | [-201.75; -130.25] | 2.4% | 32.5% |
| Gao | 15 | 50 | | -106.34 | [-112.11; -100.56] | 92.6% | 34.2% |
| Common effect model | | | | -102.70 | [-108.25; -97.14] | 100.0% | . |
| Random effects model | | | | -91.97 | [-183.11; -0.84] | . | 100.0% |

Heterogeneity: $I^2 = 97.3\%$, $\tau^2 = 6321.7008$, $p < 0.0001$

-200  -100  0  100  200

c) Forest Plot of Total Serum Ghrelin Level Difference between AIG with Severe Gastric Atrophy and Healthy Control

| Study | Severe Atrophic | Healthy Control | Mean Difference | MD | 95%-CI | Weight (common) | Weight (random) |
|---|---|---|---|---|---|---|---|
| Panarese | 19 | 18 | | -24.75 | [ -40.58; -8.92] | 14.7% | 33.7% |
| Checchi | 13 | 51 | | -192.00 | [-229.61; -154.39] | 2.6% | 32.3% |
| Gao | 13 | 50 | | -118.64 | [-125.32; -111.95] | 82.7% | 34.0% |
| Common effect model | | | | -106.72 | [-112.80; -100.65] | 100.0% | . |
| Random effects model | | | | -110.67 | [-204.77; -16.56] | . | 100.0% |

Heterogeneity: $I^2 = 98.5\%$, $\tau^2 = 6771.0013$, $p < 0.0001$

-200  -100  0  100  200

**Fig 5. Forest Plot of the Difference in Total Serum Ghrelin of (a) AIG with Mild Gastric Atrophy and Healthy Control, (b) AIG with Moderate Gastric Atrophy and Healthy Control, and (c) AIG with Severe Gastric Atrophy and Healthy Control.**

## Discussion

Three studies were included in a meta-analysis by Panarese et al. (2020), Campana et al. (2007), and Alonso et al. (2007), reported no significant difference in mean total serum ghrelin levels between patients with autoimmune gastritis and healthy controls [23–25]. Of the five studies included in the meta-analysis, only two (Panarese et al. [2020] and Alonso et al. [2007]) reported increased total serum ghrelin levels in the AIG group. Overall, the meta-analysis showed no significant difference in total serum ghrelin levels between patients with AIG and healthy controls, with a pooled mean difference of −65.28 pmol/L (five studies, 311 subjects; 95% CI: −146.05 to 15.48) (**Fig 3**). However, the heterogeneity among the studies was very high ($I^2 = 98.76\%$, $H^2 = 80.65$, Q (df = 4) = 80.65, $P < 0.0001$). Despite the overall non-significant finding, subgroup analyses based on histological grades of gastric atrophy (mild, moderate, and severe) revealed a potential severity-related trend, with significantly lower serum ghrelin levels observed in patients with moderate to severe atrophy (**Fig 4**).

The decrease in total serum ghrelin between AIG patients and healthy controls was supported by two of the included studies (Checchi et al. and Gao et al.) [26,27]. In addition, a report by Kalkan Ç, et al. (2016) found that total serum ghrelin was decreased in patients with AIG, especially those who had delayed gastric emptying [17]. These findings are consistent with the pathogenesis of AIG. Ghrelin is a peptide hormone produced by ghrelin-producing cells that acts as a ligand for the growth hormone secretagogue receptor and plays a key role in gastrointestinal regulation—stimulating gastric acid secretion, enhancing gut motility, and increasing appetite and food intake [28,29]. A key feature of AIG is the destruction of parietal cells via the recognition of proton-pump $H^+/K^+$-ATPase by APCA, which leads to gastric mucosal atrophy [3,6,5]. Ghrelin-producing cells showed close linkage by immunohistochemical analysis [30], and close proximity to parietal cells (with some cells being in full contact) [31–33]. The inflammation and gastric atrophy resulting from the destruction of parietal cells also causes the loss of ghrelin-producing cells [34], predominantly those located in the corpus (predilection of autoimmune gastritis lesions by the corpus of the stomach) [1,35] compared with other regions of the gastrointestinal tract [31].

Several included studies were not excluding the patient with *H. pylori* infection. However, because the AIG occurrence may be associated with *H. pylori* infection (concurrently or post-eradication period), the result is still valid for interpretation. In addition, this result showed that the serum ghrelin level may also associate with the condition of AIG and *H. pylori* infection that occurred concurrently. In the concurrent condition, the AIG was developed by molecular mimicry between *H. pylori* antigens and the $H^+/K^+$-ATPase with the involvement of TLR-dependent pathway [36]. This was supported by recent finding in Indonesia that showed that the positivity of parietal cell antibody (PCA) is significantly correlated to the positivity of *H. pylori* infection [20]. If the inflammation process persist after *H. pylori* eradication, the progression of AIG may occurred based on the gastric mucosal condition. However, the report of this condition is still limited to case reports [36].

Subgroup analysis findings suggest that total serum ghrelin may serve as a potential diagnostic marker for AIG, particularly in patients with moderate-to-severe gastric atrophy. However, these results were influenced by two studies with a high risk of bias and were accompanied by substantial heterogeneity. The ROBINS-E tool identified two studies, Panarese et al. (2020) and Checchi et al. (2007), as having a high risk of bias, primarily due to inadequate control of confounding variables, possible misclassification of exposure, and incomplete reporting of the outcome data. Alonso et al. (2007) were assessed as having some concerns, whereas the remaining two studies were judged to have a low risk of bias. Sensitivity analysis excluding high-risk studies was conducted to evaluate the robustness of the findings (S1 Fig). The pooled mean difference in total serum ghrelin based on the remaining three studies was −52.51 pmol/L (95% CI: −126.34 to 21.30), which remained statistically non-significant and consistent with the primary meta-analysis. This finding suggests that the main findings were not solely driven by the inclusion of studies with a high risk of bias.

The high heterogeneity observed in the meta-analysis may be attributed to the small sample size and variations in participant demographics that may affect the effect size. A report by Levine T, et al. (2009) showed that the negative correlation between the sample size and effect size which lead to overestimation [37]. To explore potential sources of

heterogeneity, a meta-regression analysis was conducted to the available possible confounding data: (1) age and (2) body mass index (BMI). The results indicated that mean BMI, but not mean age, was significantly associated with heterogeneity in pooled effect size (S2 and S3 Tables). However, the other possible confounding factors may also be contributed. Several reports in Japan showed that plasma ghrelin level was inversely correlated with metabolic syndrome and or insulin resistance [38–40]. Female gender and high-density lipoprotein level were associated independently to the ghrelin level. [40] Another report showed that plasma ghrelin was positively correlated with serum albumin level [41]. In addition, the demographic status may also contribute. Matsunaga-Irie S, et al. (2007) reported that the serum ghrelin levels were significantly higher in Caucasian than Japanese that may cause by the higher prevalence of obesity among Caucasian than Japanese [42]. Unfortunately, the limitation of provided data was limiting the exploration of other confounding factors that cause the high heterogeneity.

This study had several limitations. First, most of the included studies were conducted in Europe, with only one study from Asia (China) having a relatively small sample size, which reduces its generalizability. Second, substantial heterogeneity was observed, largely owing to differences in patient BMI. Based on these limitations, recommendations for future research include: (1) investigating serum ghrelin levels in populations from diverse geographic regions; (2) stratifying study participants based on *H. pylori* infection status to better evaluate its impact on serum ghrelin; and (3) controlling for BMI as part of the study inclusion criteria. Another potential limitation is the co-occurrence of AIG and *H. pylori* infections. However, recent publications have suggested a possible relationship between *H. pylori* infection and the progression of autoimmune gastritis [20]. Alonso et al. (2007) includes the patients with co-occurrence of *H. pylori* infection and AIG in their analysis [25]. Several previous studies have shown that 20%–30% of the *H. pylori*-infected population possess autoantibodies against parietal cells [43,44]. The fundamental mechanism is thought to involve molecular mimicry. [45] A previous study proposed that autoreactive antibodies, through the activity of $H^+/K^+$ ATPase, and *H. pylori* antigen were important contributors. This also supported by Borén et al. [46], who reported that anti-Le$^x$ monoclonal antibodies (antibody to Lewis antigens found in the LPS of *H. pylori* and human cells such as gastric epithelial cells, endothelial cells, and polymorphonuclear cells) are induced by *H. pylori* infection [47]. The β-subunit of urease in *H. pylori* is homologous to the β-subunit of ATPase in parietal cells, which also indicates a mechanism of molecular mimicry [48,49]. Therefore, *H. pylori* infection may be related to AIG in this manner.

## Conclusion

The total serum ghrelin levels in patients with AIG were not significantly different from those in healthy controls. However, patients with AIG and moderate-to-severe gastric atrophy exhibited notably lower ghrelin levels. These findings suggest that total serum ghrelin level may serve as a potential diagnostic marker for AIG in patients with advanced gastric atrophy. However, given the limited number of subjects and included studies, the applicability of these findings in clinical practice remains restricted. Given the high heterogeneity, risk of bias, and other limitations among the included studies, further well-designed research is needed to validate the diagnostic utility of serum ghrelin for AIG.

### Registration and protocol

The study protocol was not registered with PROSPERO, as data synthesis had already commenced following a preliminary literature search. This study was conducted according to Preferred Reporting Items for Systematic reviews and Meta-analyses (PRISMA).

### Supporting information

**S1 Table. Search strategy.**
(DOCX)

**S2 Table. Meta-regression analysis of mean age between studies.**
(DOCX)

**S3 Table. Meta-regression Analysis of Mean BMI between studies.**
(DOCX)

**S4 Table. PRISMA 2020 checklist.**
(DOCX)

**S1 Fig. Sensitivity analysis.**
(DOCX)

## Acknowledgments

We thank Edanz (https://jp.edanz.com/ac) for editing a draft of this manuscript.

## Author contributions

**Conceptualization:** Iqbal Taufiqqurrachman, Ari Fahrial Syam, Yoshio Yamaoka.

**Data curation:** Iqbal Taufiqqurrachman, Andro Pramana Witarto.

**Formal analysis:** Iqbal Taufiqqurrachman, Andro Pramana Witarto.

**Funding acquisition:** Yoshio Yamaoka.

**Investigation:** Iqbal Taufiqqurrachman, Murdani Abdullah.

**Methodology:** Iqbal Taufiqqurrachman, Andro Pramana Witarto, Ari Fahrial Syam, Murdani Abdullah, Sigit Ari Saputro, Irine Normalina, Muhammad Miftahussurur.

**Project administration:** Iqbal Taufiqqurrachman.

**Resources:** Yoshio Yamaoka.

**Software:** Iqbal Taufiqqurrachman, Andro Pramana Witarto, Sigit Ari Saputro.

**Supervision:** Ari Fahrial Syam, Murdani Abdullah, Muhammad Miftahussurur.

**Validation:** Iqbal Taufiqqurrachman, Ari Fahrial Syam, Yoshio Yamaoka.

**Visualization:** Iqbal Taufiqqurrachman, Andro Pramana Witarto.

**Writing – original draft:** Iqbal Taufiqqurrachman, Andro Pramana Witarto, Murdani Abdullah, Sigit Ari Saputro, Irine Normalina, Yoshio Yamaoka.

**Writing – review & editing:** Iqbal Taufiqqurrachman, Ari Fahrial Syam, Sigit Ari Saputro, Muhammad Miftahussurur, Yoshio Yamaoka.

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
