## [Decision Letter · Decision Letter 0]

23 Nov 2025

Dear Dr. Yamaoka,

Thank you for submitting your manuscript to PLOS ONE. After careful consideration, we feel that it has merit but does not fully meet PLOS ONE’s publication criteria as it currently stands. Therefore, we invite you to submit a revised version of the manuscript that addresses the points raised during the review process.

We look forward to receiving your revised manuscript.

Kind regards,

Muhammad Salman Bashir, M.S.C

Academic Editor

PLOS ONE

**Journal Requirements:**

“This report is based on work supported in part by grants from the National Institutes of Health (DK62813) (Y.Y) and Grants-in-Aid for Scientific Research from the Ministry of Education, Culture, Sports, Science, and Technology (MEXT) of Japan (221S0002, 21H00346, 22H02871, and 23K24133) (Y.Y). This work was also supported by the Japan Agency for Medical Research and Development (AMED) [e-ASIA JRP; 23jm0210090h0003, Adopting Sustainable Partnerships for Innovative Research Ecosystem (ASPIRE); 23jf0126006h0001, Science and Technology Research Partnership for Sustainable Development (SATREPS); 23jm0110025h0003], and the Japan International Cooperation Agency (JICA) [SATREPS] (Y.Y). IT is a Ph.D. student supported by the Japanese Government (MEXT) Scholarship Program for 2024.”

5. Please include your tables as part of your main manuscript and remove the individual files. Please note that supplementary tables (should remain/ be uploaded) as separate "supporting information" files.

6. Please include a copy of Table 2 which you refer to in your text on page 8.

Reviewers' comments:

Reviewer's Responses to Questions

**Comments to the Author**

1. Is the manuscript technically sound, and do the data support the conclusions?

Reviewer #1: Yes

Reviewer #2: Partly

Reviewer #3: Yes

2. Has the statistical analysis been performed appropriately and rigorously?

Reviewer #1: Yes

Reviewer #2: Yes

Reviewer #3: No

3. Have the authors made all data underlying the findings in their manuscript fully available?

Reviewer #1: Yes

Reviewer #2: Yes

Reviewer #3: Yes

4. Is the manuscript presented in an intelligible fashion and written in standard English?

Reviewer #1: Yes

Reviewer #2: Yes

Reviewer #3: Yes

Reviewer #1: The manuscript is well written, and the topic is highly relevant, especially given the diagnostic challenges of autoimmune gastritis (AIG) and the growing interest in non-invasive serum biomarkers. However, there are significant methodological limitations that weaken the conclusions. The study included only five articles, two of which had a high risk of bias. Additionally, several points require improvement and more thorough discussion. Overall, this is a promising study on an important topic.

Major issues:

The authors include studies with AIG patients both with and without Helicobacter pylori infection. However, the introduction emphasizes that H. pylori confounds the clinical picture. This issue should be addressed more thoroughly by the authors throughout the discussion.

The study's heterogeneity is very high, and the discussion of its relevance is insufficient. The result for the mean difference in total serum ghrelin between AIG patients and healthy controls was accompanied by high heterogeneity (I² = 98%). This level of I² indicates that the observed variation in the results was not random. All of this should be explored further in the discussion. Discuss in more detail the reasons for the high heterogeneity and what this implies for the results.

The conclusions suggest the diagnostic utility of ghrelin in cases of moderate or severe gastric atrophy; however, this conclusion should be interpreted with greater caution. I recommend moderating the conclusion and highlighting the study's limitations. Although subgroup analysis provides valuable insights, the main conclusion should be based on the pooled outcome. Given the heterogeneity and bias in the analyses, the authors should exercise more caution and use more reserved language when interpreting the clinical significance of the subgroup findings, clearly stating that these findings lack robust validation.

Minor issues:

The authors state that they did not register the systematic review protocol in PROSPERO before data extraction. For current systematic reviews, the lack of registration is a methodological limitation that should be discussed rather than simply mentioned. Please include a section justifying the impact of this omission and discuss how methodological decisions were pre-defined.

Although four databases are mentioned, there is inconsistency in the list of databases. The Methods section lists PubMed, EBSCO Host, Scopus, and ScienceDirect, while the Results section mentions PubMed, EBSCO Host, Scopus, and ProQuest. The authors should correct the list to accurately reflect the databases used.

The search period is stated as "August 25 to September 31, 2024." September does not have 31 days, so this error should be corrected.

Reviewer #2: Regarding the appreciated efforts of the authors in preparing this meta-analysis review by addressing the possible use of Ghrelin as indicative biomarker of AIG, there are some important notes should be discussed first:

Introduction section:

lines 86-89 lack the proper peer reviewed citations to support authors claims regarding AIG low incidence in Asian countries. Proper references can be: “5”, “Terao, S., Suzuki, S., Yaita, H., Kurahara, K., Shunto, J., Furuta, T., Maruyama, Y., Ito, M., Kamada, T., Aoki, R., Inoue, K., Manabe, N., & Haruma, K. (2020). Multicenter study of autoimmune gastritis in Japan: Clinical and endoscopic characteristics. Digestive endoscopy : official journal of the Japan Gastroenterological Endoscopy Society, 32(3), 364–372. https://doi.org/10.1111/den.13500“, and “Park JY, Cornish TC, Lam-Himlin D, Shi C, Montgomery E. Gastric lesions in patients with autoimmune metaplastic atrophic gastritis (AMAG) in a tertiary care setting. Am J Surg Pathol. 2010;34:1591–1598. doi: 10.1097/PAS.0b013e3181f623af.” “Notsu T, Adachi K, Mishiro T, et al. Prevalence of autoimmune gastritis in individuals undergoing medical checkups in Japan. Intern Med. 2019;58:1817–1823. doi: 10.2169/internalmedicine.2292-18.”

Suggested references (lines 97-99):

- Park, J. Y., Cornish, T. C., Lam-Himlin, D., Shi, C., & Montgomery, E. (2010). Gastric lesions in patients with autoimmune metaplastic atrophic gastritis (AMAG) in a tertiary care setting. The American Journal of Surgical Pathology, 34(11), 1591–1598. https://doi.org/10.1097/pas.0b013e3181f623af

- Rugge, M., Bricca, L., Guzzinati, S., Sacchi, D., Pizzi, M., Savarino, E., Farinati, F., Zorzi, M., Fassan, M., Tos, A. P. D., Malfertheiner, P., Genta, R. M., & Graham, D. Y. (2022). Autoimmune gastritis: long-term natural history in naïveHelicobacter pylori-negative patients. Gut, 72(1), 30–38. https://doi.org/10.1136/gutjnl-2022-327827.

- Yu, Y., Tong, K., Shangguan, X., Yang, X., Wu, J., Hu, G., Yu, R., & Tan, C. (2023). Research status and hotspots of autoimmune gastritis: A bibliometric analysis. World Journal of Gastroenterology, 29(42), 5781–5799. https://doi.org/10.3748/wjg.v29.i42.5781

- Fuentes-Valenzuela, E., Cruz, S. E., Villanueva, J. P., Rodríguez, A. C., Díaz, A. G., Del Carmen López-Martín, M., De La Plaza, I. R., Blanco, S., Martín-Falquina, I. C., Aran, B. R., Martinez, R. L., Herrera, L. a. C., Olvera, R., Rodríguez, D. A., López, K. G., & Domínguez, A. B. (2025). Prevalence and evolution of newly diagnosed autoimmune gastritis in a large Spanish retrospective cohort. Revista Española De Enfermedades Digestivas, 117(8), 441–446. https://doi.org/10.17235/reed.2025.11101/2025

Lines (103-106): Since this section describes the correlation between mucosal atrophy and declined serum ghrelin levels, the Inverse correlation between gastrin and histologic severity, and what are the functional implications that can be contributed with reduced ghrelin levels should be declared to support the authors’ hypothesis in Ghrelin use as diagnostic biomarker. (suggested references:

- Checchi, S., Montanaro, A., Pasqui, L., Ciuoli, C., Cevenini, G., Sestini, F., Fioravanti, C., & Pacini, F. (2007). Serum Ghrelin as a Marker of Atrophic Body Gastritis in Patients with Parietal Cell Antibodies. The Journal of Clinical Endocrinology & Metabolism, 92(11), 4346–4351. https://doi.org/10.1210/jc.2007-0988

- Kalkan, Ç., & Soykan, I. (2018). The relations among serum ghrelin, motilin and gastric emptying and autonomic function in autoimmune gastritis. The American Journal of the Medical Sciences, 355(5), 428–433. https://doi.org/10.1016/j.amjms.2017.12.021

- Wu W, Zhu L, Dou Z, Hou Q, Wang S, Yuan Z, Li B. Ghrelin in Focus: Dissecting Its Critical Roles in Gastrointestinal Pathologies and Therapies. Current Issues in Molecular Biology. 2024; 46(1):948-964. https://doi.org/10.3390/cimb46010061

Additional suggestion regarding INTRODUCTION SECTION support:

Conceptual diagrams showing the relationship between AIG, ghrelin, and gastric atrophy to address the introduction concepts must be provided (by using visualization tools such as Canva or BioRender).

Methods

Search strategy: By referring to results section and supplementary table 2, the fourth database must be corrected as ProQuest and authors must pay attention to information integrity, consistency and accuracy prior to any future submission!

Selection and Data Collection Process: I believe that Systematic Review Management is more appropriate to be for better handling of study selection and bias assessment (such as Covidence, Zotero or Rayyan), which can also be reflected in the discussion. Also, the avoidance of using any automated / AI tools for data collection should be explained and define the reasons behind applying the manual data collection instead!

Results:

Study characteristics: This must be addressed in a summary (at least one to two paragraphs) along with the mentioned table 1; by illustrating the main parameters being mentioned, methods of Ghrelin measurement being used, and focus on 1/5th presentation of Asian countries among the included studies!

Discussion section:

The study includes a meta-analysis of five studies with subgroup analyses based on atrophy severity, enhancing the depth of interpretation. These analyses strengthen the robustness of the findings by exploring heterogeneity and bias sources and clarified the results transparency through well reporting of effect sizes, confidence intervals, and heterogeneity metrics (I², Q, H²).

- Despite meta-regression, the I² remains very high (>98%), and the discussion could better explore other sources (e.g., assay methods, geographic variation).

- As we’ve noticed before (in baseline characteristics table), that most studies are from Europe, with only one from Asia, limiting generalizability, and thus deep analysis should explain the studies limitation. Moreover, BMI is identified as a contributor to heterogeneity, but other potential confounders (e.g., age, comorbidities, medication use) are not explored.

- The discussion acknowledges overreliance on small sample sizes but could better emphasize how it affects statistical power and interpretation.

Reviewer #3: The study addresses a relevant clinical question regarding non-invasive diagnostics. However, the conclusions are heavily limited by the very high heterogeneity and the small number of available studies.

-Major Weaknesses and Limitations

1.  Very High Heterogeneity: This is the most critical limitation of the meta-analysis. An `I²` value of 98% indicates that almost all the variation in results across studies is due to heterogeneity. The authors acknowledge this but the results should be interpreted with extreme caution.

2.  Small Number of Included Studies: With only five studies, the analysis has limited statistical power.

3. The authors note that most studies were from Europe, with only one from Asia, and sample sizes were generally small. This limits the generalizability of the findings.

**Do you want your identity to be public for this peer review?** For information about this choice, including consent withdrawal, please see our Privacy Policy

Reviewer #1: No

Reviewer #2: No

Reviewer #3: **Yes:** Mohamed Saad Hashim

---

## [Author Response · Author response to Decision Letter 1]

15 Dec 2025

Dear Reviewers,

Thank you very much for your thoughtful and constructive comments on our manuscript. We have revised the manuscript thoroughly in accordance with the suggestions provided by both the Editors and the reviewers. Despite certain limitations related to the available data, we have addressed all points to the best of our ability and made substantial improvements to enhance the clarity, accuracy, and rigor of the study.

A detailed point-by-point response to each comment has been uploaded in the "Response to Reviewers" file.

We sincerely appreciate the time and expertise the reviewers devoted to evaluating our work.

Best regards,

Yoshio Yamaoka, M.D., Ph.D

---

## [Decision Letter · Decision Letter 1]

16 Feb 2026

Diagnostic potential of total serum ghrelin in autoimmune gastritis: a systematic review and meta-analysis

PONE-D-25-40981R1

Dear Dr. Yamaoka,

We’re pleased to inform you that your manuscript has been judged scientifically suitable for publication and will be formally accepted for publication once it meets all outstanding technical requirements.

Kind regards,

Chen Ling, Ph.D.

Academic Editor

PLOS One

Additional Editor Comments (optional):

Reviewers' comments:

Reviewer's Responses to Questions

**Comments to the Author**

Reviewer #1: All comments have been addressed

Reviewer #2: All comments have been addressed

Reviewer #3: All comments have been addressed

2. Is the manuscript technically sound, and do the data support the conclusions?

Reviewer #1: Yes

Reviewer #2: Yes

Reviewer #3: Partly

3. Has the statistical analysis been performed appropriately and rigorously?

Reviewer #1: Yes

Reviewer #2: Yes

Reviewer #3: Yes

4. Have the authors made all data underlying the findings in their manuscript fully available?

Reviewer #1: Yes

Reviewer #2: Yes

Reviewer #3: Yes

5. Is the manuscript presented in an intelligible fashion and written in standard English?

Reviewer #1: Yes

Reviewer #2: Yes

Reviewer #3: Yes

Reviewer #1: (No Response)

Reviewer #2: We would like to thank you for giving me the opportunity to review the revised version of the manuscript entitled "Diagnostic potential of total serum ghrelin in autoimmune gastritis: a systematic review and meta-analysis," in which the authors worked extensively on modifying their manuscript in response to the previous peer reviewers' commentaries:

1- Editor's notes regarding grant information concealment and updating the financial disclosure in the manuscript's body, as well as the corrections in the supplementary files.

2- Reviewers' comments regarding:

— Providing a detailed explanation of the interaction between H. pylori infection and AIG in the discussion part.

— Discussing the attribution of the high heterogeneity with the small sample size and how the variations in participant demographics might affect the effect size.

— Concluding the restricted clinical applicability of the study and also moderating the interpretation of the subgroup analyses accordingly to not be overgeneralized.

— Updating of certain manuscript sections through incorporating some of the recommended references by the peer reviewers to strengthen the authors' prepositions.

Reviewer #3: The authors responded to the reviewers comments and acknowledged the limitatins and shortcoming of the study.

**Do you want your identity to be public for this peer review?** For information about this choice, including consent withdrawal, please see our Privacy Policy

Reviewer #1: No

Reviewer #2: No

Reviewer #3: **Yes:** Mohamed Saad Hashim

---

## [Editor Report · Acceptance letter]

PONE-D-25-40981R1

PLOS One

Dear Dr. Yamaoka,

I'm pleased to inform you that your manuscript has been deemed suitable for publication in PLOS One. Congratulations! Your manuscript is now being handed over to our production team.

Kind regards,

on behalf of

Dr. Chen Ling

Academic Editor

PLOS One